# ‘Organ’ising Floral Organ Development

**DOI:** 10.3390/plants13121595

**Published:** 2024-06-08

**Authors:** Kestrel A. Maio, Laila Moubayidin

**Affiliations:** Department of Cell and Developmental Biology, John Innes Centre, Colney Lane, Norwich NR4 7UH, UK; kestrel.maio@jic.ac.uk

**Keywords:** flower organs, symmetry, identity, axiality, gynoecium, homeobox, hormones, flowers

## Abstract

Flowers are plant structures characteristic of the phylum Angiosperms composed of organs thought to have emerged from homologous structures to leaves in order to specialize in a distinctive function: reproduction. Symmetric shapes, colours, and scents all play important functional roles in flower biology. The evolution of flower symmetry and the morphology of individual flower parts (sepals, petals, stamens, and carpels) has significantly contributed to the diversity of reproductive strategies across flowering plant species. This diversity facilitates attractiveness for pollination, protection of gametes, efficient fertilization, and seed production. Symmetry, the establishment of body axes, and fate determination are tightly linked. The complex genetic networks underlying the establishment of organ, tissue, and cellular identity, as well as the growth regulators acting across the body axes, are steadily being elucidated in the field. In this review, we summarise the wealth of research already at our fingertips to begin weaving together how separate processes involved in specifying organ identity within the flower may interact, providing a functional perspective on how identity determination and axial regulation may be coordinated to inform symmetrical floral organ structures.

## 1. Introduction

The current understanding of flower morphogenesis is fragmented and must be coordinated to progress the field of plant biology and for societal progression in the face of rising temperatures and increasing pressure on crop production. Flowers are reproductive structures; we rely on them to propagate many of our crop species as well as to produce the grains, seeds and fruits we directly consume. Improving our understanding of these structures may allow for controlled manipulation to overcome impending hunger crises as food production struggles to keep up with a fluctuating climate and the demands of a growing population. Animals, including insects and birds, pollinate 87.5% of flowers. As a result, their shape, smell, and aesthetic are integral to their propagation as well as their function [1]. Humans have been attempting to study the structure of flowers since the Greek philosopher Theophrastus, described as the father of botanical science, began to define the structure and roles of the ephemeral parts of the flower [2]. Our understanding has evolved greatly since then, but we still have no straightforward answer to the question: how do complex symmetric shapes develop from a shapeless mass of undifferentiated tissue?

This ‘shapeless mass of undifferentiated tissue’ is termed inflorescence primordia (IM) (Figure 1A). Through phyllotactic patterning, a subset of undifferentiated cells in this IM will ‘bud-off’ and develop a floral meristem (FM) (Figure 1B), which is what then develops into the flower [3,4]. In the model organism *Arabidopsis thaliana*, the position of new established flowers adheres, roughly, to the ‘golden angle’ of 137.5° [5] (Figure 1B). This is genetically directed, in part, by *LEAFY* (*LFY*) and *APELATA1* (*AP1*) in *Arabidopsis thaliana*, and by the orthologous *FLORICAULA* (*FLO*) and *SQUAMOSA* (*SQUA*) in the model organism *Antirrhinum majus* [4,6,7]. From here, there is a complicated network of genetic interactions specifying identity, positioning, growth, and axis, as well as hormonal homeostasis leading to signalling outputs which function to commit groups of cells into organs. The flower organs are organised into whorls (concentric circles) or spirals. In the model organisms *Arabidopsis thaliana* and *Antirrhinum majus*, four concentric whorls, considered here as grouping the organs themselves, are specified by a set of key regulators and their genetic interactions. These genes are conserved and well-documented [8]. However, they do not account for the whole picture. Growth in an immobile system is reliant on the implementation of genetic and environmental signals, which can influence the determination of body axes in order to shape the organs. In other model organisms, such as *Drosophila*, a dynamic combination of spatiotemporal controlled gene patterning and morphogen gradients links the anterior–posterior positioning of the body axes to fate determination and bilateral symmetry [9,10,11,12]. Whether the same basic rules could be applied in the formation of a body plan and establishment of shape in the plant kingdom will be explored in this review. The primary question of how the processes of body axis development, symmetrical organisation, and identity establishment interact with each other to shape a plant organ will be addressed. Further, the possibility that signals equivalent to morphogens in plants could aid communication between these processes will be discussed.

## 2. Symmetry

### 2.1. The Beauty of Biological Symmetry

Living systems often exploit geometry to optimise shape, create new functions, and minimise energy expenditure. Symmetry refers to the divisions into smaller, identical parts that an object can be subject to. In physics, art, and mathematics, symmetry is often considered a basal state and as such, breaks in its regularity and uniformity reveal disruptions to a system. Breaks in symmetry form the basis for plays and stories and are often explored in the visual arts. The human eye and mind is fascinated by this change in regularity in a system [13,14]. However, the converse is also true. The golden mean has been described since the inception of Greek architecture to be the division and structure of columns within the Parthenon most satisfactory to our understanding of aesthetics. This ‘golden mean’ has since been connected to Fibonnacci numbers and plant growth, describing phyllotaxis, a regular arrangement of lateral organs along the aerial part of the plant [15]. Within biology and human psychology, symmetry is considered desirable, in terms of attracting mates (for animals) and pollinators (for plants); hence, it can be considered a factor of reproductive success [16,17,18]. Plants are reliant on genetically pre-determined floral structures to allow for successful reproduction and species propagation. Flower symmetry is both pleasing to the eye as well as functional. The shape of flowers enhances the efficiency of pollination, with bees showing an innate preference for bilaterally symmetric flowers, hence its impact on reproductive success [19]. Another study by Helversen et al. [20] demonstrated that the bell shape of flowers pollinated by bats allowed for enhanced pollination due to their comparatively long echoes, allowing bats to locate them more easily. The shape of flowers and their colour patterning, both of which can be symmetrical, is also an important consideration when looking at human psychology. The aesthetic of flowers, fruits, and vegetables impacts consumer choice, and has been investigated from a psychological perspective in order to better understand how to minimise the pervasive issue of food waste [21]. 

Molecular studies have reconstructed phylogenies, which have revealed that ancestral flowers displayed a radial symmetry, often described as actinomorphy, which essentially relates to an object which can be divided into identical parts along a longitudinal plane across its central axis [22]. Since, floral species have undergone mass diversification, not only in shape but also in axis symmetry. *Antirrhinum majus* is a well-established and studied example of bilateral symmetry, in which the dorsal axis of the flower exhibits a different conformation to the ventral axis, and therefore cannot be divided across axes in a radial manner [23,24]. Flowers, such as *Chamaelirium japonicum*, can also mimic a bilateral symmetry where they are, in fact, asymmetric due to differential tepal growth [25].

### 2.2. Symmetry Establishment in Plants

Regarding the establishment of the structure and symmetry of inflorescences themselves, two types of model flowers, *Arabidopsis* and *Antirrhinum* (Figure 2), are studied extensively. As previously mentioned, the *Antirrhinum* flower exhibits bilateral symmetry according to its arrangement of floral organs. This bilaterality is predominantly due to the differing growth patterns of the stamens and the petals. The genes *CYCLOIDEA* (*CYC*), *DIVACARTA* (*DIV*), and *DICHOTOMA* (*DICH*) have been identified as major genetic players in defining bilateral symmetry within *Antirrhinum* flowers via the regulation of growth rates [23,26,27,28]. These genes act in similar ways in other plant species. In the close relative of *Antirrhinum*, *Mohavea confertiflora* (the desert flower), the corolla shape has shifted with a superficial radial symmetry, due to alterations in *CYC* and *DICH* expression [29,30,31]. Orthologues of *CYC* have also been described in other floral species, such as the toadflax flower (*Linaria*) [32], and to be key to shifts in symmetry in *Proteaceae*, a basal eudicot family with high amounts of variation in symmetry [33].

Symmetry establishment in terms of floral organ arrangement can be elucidated by studying the gene product *PERANTHIA* (*PAN*), a b-ZIP transcription factor that when mutated leads to pentameric (5-fold) radial symmetry on the flower without affecting the size of the IM, meaning that the supernumerary flower organs are initiated from the same given space and number of cells at time of initiation [34]. Pentamery represents the ancestral state, as it has been identified as a frequent and basic form throughout the sub-classes of *Rosidae* and *Asteridae* [35,36,37]. Therefore, it has been suggested that the tetramerous (2-fold, bi-radial) flowers of the *Brassicaceae* family are due to the acquisition of PAN function, which maintains the correct patterning of the abaxial and adaxial sepal founder cell populations [38] (Figure 2A). Initiation and distinction of identity of floral organs occur separately [39]. It is therefore important to understand the processes tying these two together—at what point does identity come into play in shaping the organ? The positioning of floral organs has been modelled to be a consequence of inhibitory dorsal effects on ventral primordial development [40]. For instance, *CYC* acts dorsally to inhibit ventral identity in *Antirrhinum* (Figure 2A). In the loss-of-function *cyc* mutant, the dorsal petal identity is partially lost while ventral (and lateral) identity expands dorsally [23]. This is due to both main axis size and inhibitory effects. This model holds true in *Arabidopsis*, where symmetrical initiation of organs is also dependent on ventral inhibition by genes such as *PAN*, *LFY*, *BLADE-ON-PETIOLE1* (*BOP1*), and *BOP2* (Figure 2A), as well as the position of the bract [40]. Presumably, there must be some factor(s) integrating the position of an organ with its identity. Therefore, it is necessary to understand the coordination of symmetry in terms of the positioning of organs and initiations, and this can be partially described through the action of hormones, as explained below (Figure 2B).

### 2.3. Morphogens and Phytohormones in Plant Symmetry Establishment

In *Drosophila*, the body plan is established by the antagonism of morphogens acting at opposite poles of the developing organisms [41,42]. The term morphogen here describes a biochemical substance capable of diffusing through cells with the end result being a response to its presence [43]. This term was coined in 1952 by Alan Turing [44]. In plants, the phytohormone auxin can be considered to act as a morphogen as it alters the position and rate of growth of organs during development [45,46]. Furthermore, in a similar manner to how animal morphogens control the development of organs in a dose-dependent manner, auxin also acts along a gradient [45,47,48]. Auxin has, been shown to pattern the xylem distribution in a concentration-dependent manner, distinguishing division, expansion, and secondary wall formation [49]. This phytohormone can also manipulate floral symmetry. For instance, in *Antirrhinum*, floral buds exposed to localised auxin application formed radially symmetric flowers as opposed to their typical bilateral conformation [50]. Auxin is also crucial in the determination of organ initiation in flowers and thus drives the tetrameric symmetry of *Arabidopsis*. Its activity is localised to sites of organ initiation, specifying founder cell sites, which direct where floral organs are positioned in the flower [51]. Auxin has been demonstrated to act in the initiation of petals and sepals in the flower, specifying their initiation sites from the epidermis of the floral meristem [52,53]. To facilitate the outgrowth of sepals (which, as shown in Figure 2B, arise alternately to petals) auxin participates in cross-talk with another phytohormone, cytokinin. There is a well-documented dichotomy in the function of these two phytohormones, with a decrease in auxin delaying the initiation of floral organs and vice versa with an increase in cytokinin [53]. The gene *LEAFY* (*LFY*), which, as previously discussed, plays an integral role in the development of the biradial symmetry of *Arabidopsis* flowers, interacts with auxin to determine the position and initiation of organs in developing flowers [40,54]. Similarly, the transport and accumulation of auxin has been shown to drive transitions in symmetry at the gynoecium apex, switching from a bilateral ovary (at the base of the gynoecium) towards a radial style (at the top of the gynoecium) through apolar distribution of the auxin transporters PINs, which in turn facilitate the progressive establishment of an auxin ring [55].

### 2.4. Transitions in Flowers and Floral Organ Symmetry

Transitions in floral symmetry are well documented, with 199 changes in perianth symmetry identified throughout angiosperm evolution [56]. Transitions can also occur during the development of some flowers. The default condition for these organ-specific transitions tends to be from radial to bilateral symmetry. One potent example is in the *Aconitum* genus (*Ranunculaceae* family) where the actinormophic (radially symmetric) developing flower transitions towards zygomorphy (bilaterally symmetric) through the development of a petal spur on just one side of the flower [3,57,58]. Symmetry can also be linked to genes related to floral organ identity and growth in the *Fumarioideae* flower where paralogs to the *CRABSCLAW* (*CRC*) gene, a master regulator of gynoecium identity in *Arabidopsis* [59], and *CYC*, which is essential for growth control in *Antirrhinum* [23], are responsible for the shift from biradial symmetry towards a bilateral conformation throughout the development of the flower. Rarely, in plants, do radial-to-bilateral symmetry transitions occur [60]. There is a rare occurrence of this switch in the gynoecium of the *Arabidopsis* flower, where the bilateral ovary (the structure which contains ovules at the base of the gynoecium) transitions towards a radial style (the structure supporting the stigmatic papillae through which the pollen is transmitted allowing for efficient fertilisation) (Figure 3) [55,61,62,63]. bHLH and NGATHA (NGA) transcription factors play key roles in the apical closure of the gynoecium, which facilitates this switch in symmetry. These transcription factors work both in a hierarchy and in cis, between themselves. Downstream, it is apparent that the bHLH transcription factors regulate the transport of auxin, whilst the NGAs control auxin biosynthesis to facilitate their described functions [64]. In terms of the transition towards radial symmetry, it was found that two bHLH transcription factors SPATULA (SPT) and INDEHISCENT (IND) are responsible for repressing margin identity genes, allowing the structure to develop radially at the most distal part. The transition in symmetry is driven by dynamic auxin accumulation, which is orchestrated by SPT- and IND-dependent regulation of the auxin polar transporter (PIN) distribution at the plasma membrane. Notably, apolar PIN distribution (opposite to the canonical polar distribution) allows the formation of auxin maxima, or foci, positioned apically, underlying the lateral carpels (bilateral stage), then the medial axis (bi-radial, four-foci stage), and lastly forming an auxin ring that encompasses the adaxial axis, allowing for radial symmetry establishment at the gynoecium apex (Figure 3A). These genes are sufficient to support transitions in symmetry in young leaves as ectopic expression of *IND*, in the presence of a functional copy of SPT, in the bilateral leaf of *Arabidopsis*, leads to a transition towards radial symmetry. The establishment of symmetrical shape requires careful coordination across axes. SPT/IND heterodimers coordinate the medial–lateral axis together with auxin [65]. Another important axis to consider in the establishment of symmetry is the adaxial–abaxial axis, which is regulated by two Homeodomain Leucine Zipper class-II (HD-Zip II) transcription factors: Homeobox Arabidopsis Thaliana 3 (*HAT3*), and Arabidopsis Thaliana HomeoBox4 (*ATHB4*) (Figure 3B). These genes mediate the establishment of the auxin ring from a biradial, four-foci stage and are direct downstream targets of SPT in this process [55]. Loss of these two transcription factors results in a split-style phenotype that differs from that of *spt* mutants for its diagonal position, which is due to the loss of the adaxial body axis. While the *hat3 athb4* double mutant is able to accumulate auxin in a four-foci stage supporting the development of the medial tissues, it fails to proceed from a biradial state of auxin maxima to the ring-shaped accumulation, thus specifically supporting the last step in symmetry transition at the gynoecium apex (Figure 3B) [63].

In addition, *SPT* functions as an organ identity gene, presumably acting downstream and in parallel of *AGAMOUS* (*AG*) in the developing gynoecium [59,66]. Therefore, the genetic interaction of *HAT3* and *ATHB4* homeobox genes with the organ identity gene *SPT* during style morphogenesis is the first tie, in this review, between these key developmental processes of identity and axiality.

The key components in establishing symmetry in the gynoecium have been identified genetically, *SPT*, *HAT3* and *ATHB4*, and they have ties to organ/tissue polarity and identity. Could these genes play a core, conserved role in the development of symmetry and its coordination with organ identity? In the gynoecium, we see the importance of the master regulator of carpel organ identity (*SPT*) on the formation of radial symmetry in the style of the gynoecium, and how it mediates this transition through interactions with the phytohormone auxin. Therefore, there could be both genetic and biochemical inputs regulating these two processes of establishing, in a spatio-temporal manner, identity and symmetry in plants. These principles in specifying symmetry establishment may have a conserved role in other floral organs; therefore, how these processes are linked to floral organ identity must be explored further.

## 3. Determining Organ Identity Establishment

### 3.1. Flower Identity and Distinction from Leaf-Like Lateral Organs Is Conferred by ABC Genes

The commitment and spatio-temporal regulation of flower organ identities have been studied extensively by developmental geneticists in the last 30 years, reviewed in [67]. Identity is reliant on the shape and function of the organ it describes, which gives insight to the complexity ingrained in understanding its establishment. To consider organ identities in plants, the most common model is the flower. Although still openly debated, flowers are considered leaf homologue structures, an idea first proposed back in the late 18th century by Goethe [68]. In order to confer a distinct identity to the organs defining and forming a flower, a set of genes described in a pioneering article by Coen and Meyerowitz in 1991 as *ABC* genes are required [8]. These genes are not only integral to the development of the identity and shape of the floral organs themselves, but also in distinguishing them from their assumed predecessors: the leaves. Follow-up experiments demonstrated that loss of these *ABC* genes resulted in the leaf-like development of floral organs, identifying them as core regulators in the definition of flowers [69,70,71]. Model flowers are formed of four distinct whorls which encompass four unique organ types. The definition of these whorls begins at the floral meristem (FM) stage, following differentiation from the inflorescence meristem (IM) (Figure 1). They are defined from the outermost whorl. This whorl informs the development of the sepals, which are protectors of the internal whorls and supporters of the petals. Their shape and function have curiously been linked to pollinator strategies in *Clematis stans* flower, where the length of the calyx tube, formed of four sepals, changes temporally to facilitate efficient pollination by two bumblebee species [72,73]. Petals form the often ornamental second whorl in the flower and in some instances they also host nectaries to attract pollinators. Their shape and arrangement is often specific to pollinator preference [74] and can be considered the most relevant whorl in terms of creating the overall flower shape, including its symmetry, as discussed previously. The stamens are male reproductive structures in the third whorl of the flower, which host the pollen to fertilise and propagate the plant. Finally, the female reproductive organ, the gynoecium, resides within the innermost, or fourth, whorl of the flower. This is shaped in various ways but can typically be described as an ovary, formed of one or multiple fused or unfused carpels (e.g., two congenitally fused carpels are present in *Arabidopsis*), which is topped with a style and stigmatic tissue, which chemically and biochemically interact with pollen to facilitate fertilisation of the plant (Figure 4) [75,76].

### 3.2. The ABCDE Model: An Extension of the ABC Model Describes the Designation of Floral Organ Identity

In the development of flowering organs, the organisation, here considered in terms of axiality and symmetry of these organs, must be tightly regulated. The ABC model, eventually extended to the ABCDE model, is a well-defined gene network that underpins the complexity of flower shape acquisition. It describes, essentially, how a shapeless floral primordium develops the individual and complex structures forming a flower. Each organ has a specific shape and function in the reproduction of the plant, and so its genetic regulation is similarly distinct. The model was uncovered by Coen and Meyerowitz in 1991 [8] and succinctly provides an example of homeosis in a plant system.

Three classes of genes, A, B and C, control the development of the four whorls of the flower (Figure 4). The A-class genes, *APELATA-1* (*AP1*) and *APELATA-2* (*AP2*) in *Arabidopsis* determine the development of the sepals and, in combination with B-class genes, *APELATA-3* (*AP3*) and *PISTILLATA* (*Pi*), the petals. The C-class gene, *AGAMOUS* (*AG*), is mutually repressive with A-class genes, giving them distinct expression domains within the flower. These genes specify the stamen, in combination with the B-class of genes, and the carpels, independently. These genes specify MADS-box transcription factors (with the exception of the A-class gene *APELATA-2*), which act in tetrameric complexes in order to confer specificity to the transcription of genes in particular organs and tissues [77]. The A-, B-, and C-classes of genes are relatively few; therefore, there are added layers of complexity to this model—the tetrameric complexes demonstrate how these genes can act to direct a wide range of processes through collaborative action.

The D-class genes describes other genes required in combination with the C-class genes to specify the development of the gynoecium. These genes are *SEEDSTICK* (*STK*)*, SHATTERPROOF1* (*SHP1*), also known as *AGAMOUS-LIKE 1* (*AGL1*), and *SHP2* (or *AGL5*), and are required to specify ovule identity [78]. Though *AG* is crucial for the development of the gynoecium, as the only C-class gene represented, it is also apparent that other master regulators such as *SPT* and *CRABSCLAW* (*CRC*) are required downstream in the development of this organ [59]. *SHP1,2* also act redundantly with *AG*, as their *AGL* name suggests. Where the *SHP1* and *SHP2* genes were removed in loss-of-function mutants *ap2agshp1shp2*, all evidence of ectopic carpelloid organs (homeotic conversions) forming in the outer whorls of an *ap2* mutant was removed, demonstrating that they act independently of *AG* to specify carpel identity and are repressed in outer whorls in a similar manner [79]. The E-class genes encompass the *SEPALLATA1* (*SEP1*)*, SEP2, SEP3*, and *SEP4* genes, which act mostly redundantly in tetrameric complexes with the MADS-box transcription factors of the ABC model in order to confer specificity to floral organs [80,81].

### 3.3. Diversity of the ABCDE Model to Suit Environment and Function

The ABCDE model was first identified and described in both *Arabidopsis* and *Antirrhinum*; however, they both diverge at the A-class part of the model. Evolution and gene duplications and redundancies have resulted in a varied model, which is in part defined by particular environments. The *Arabidopsis* and *Antirrhinum* homologous identity of A-class genes have remained a mystery throughout the clades, with *Arabidopsis* being one of the few models in which candidate A-class genes have been identified. AP1 has been shown in both *Arabidopsis* and *Antirrhinum* to act in specifying floral meristem identity, but in *Arabidopsis*, this MADS-box transcription factor also works to specify sepal and petal identity. However, in *Antirrhinum* this is not the case. The orthologue of the *AP1* gene is *SQUAMOSA* (*SQUA*) in *Antirrhinum*, and is required for the establishment of floral meristem identity (as it is in *Arabidopsis*), although unlike the *Arabidopsis* orthologue it is not required to establish the perianth (sepals and petals together) [82,83]. As a result of this, the ABC model is often referred to as the (A)BC model in *Antirrhinum*. *DEFICIENS* (*DEF*) and *GLOBOSA* (*GLO*) are *Antirrhinum* orthologues equivalent to *AP3* and *Pi*, respectively, and *AG* function in reproductive organ identity has been divided into two MADS-box genes *FARINELLI* (*FAR*) and *PLENA* (*PLE*) [84,85,86]. *far* mutants show reduced male fertility, whereas the *ple* mutants have sepal-like carpels. However, *AG* is orthologous with *FAR*, whereas *PLE*, with a more consistent phenotype with *ag*, shows higher sequence similarity to the D-class *SHATTERPROOF* genes [86,87]. In a similar manner to the repressive function of A-class genes on C-class genes in the canonical ABC model, the genes *STYLOSA* (*STY*) and *FISTULATA* (*FIS*) suppress C-class function, also. Mutants of these genes show an expansion of the C-class gene *PLE*, as well as carpelloid sepals; however, they appear to do so through hierarchical control of downstream genes interacting with the B- and C-class genes, or gene regulators such as *FIMBRIATA* (*FIM*)*, CHORIPETALA* (*CHO*), and *DESPENTEADO* (*DESP*) [88,89].

Rice (*Oryza sativa*) is a monocotyledonous flower and, therefore, is distantly related to *Arabidopsis* [90,91]. Rice, in contrast to *Arabidopsis*, does not possess four canonical whorls. Instead, it shares a typical conformation of whorls to other monocotyledonous flowers, with organs arranged into five whorls [92]. Rice flowers possess a gynoecium, which is branched at its style region (unlike in *Arabidopsis*), six stamen (divided into inner and outer), and two lodicules enclosed by leaf-like structures, the lemma and palea, and altogether these form a floret. The lodicules are glandular organs which swell to push the lemma and palea apart to facilitate wind pollination [92,93]. Two rice genes with a conserved AG-like function are OsMADS3 and OsMADS58, which appear to have sub-functionalised, in contrast to AG, with OsMADS3, playing a predominate role in stamen specification, and OsMADS58 (also referred to as DROOPING LEAF, or DL, as in Figure 4C) acting in carpel morphogenesis [94,95]. Therefore, whilst rice has diverged in shape and in floral organ arrangement, the conservation of this model is evident, even between these distantly related plant species.

### 3.4. Interactions with the ABCDE Model Confer Specificity at a Tissue and Organ Level

As has been previously discussed, floral organ initiation is directed, spatially, by auxin activity, whilst the identity of the organs is specified by the aforementioned ABCDE model and MADS-box transcription factors. Whilst these identity determinants distinguish organ identities from each other, they do not, independently, explain the development of floral shape and the distinction of tissues and structures within each organ. Another factor to note is that when the ABC genes were expressed in leaves, they were insufficient to induce the development of a flower in an ectopic manner, suggesting that their function in regulating flower development is reliant from other organ inputs [96]. Therefore, as has been outlined, it was previously suggested that the ABCDE classes of genes act in complexes which confer specificity to direct organ emergence and development. Essentially, this proposes that floral organs develop through specific genetic networks regulated by tetrameric complexes directed by MADS-box genes [97]. These ‘quartets’ bind to DNA *CArG* (CCArichGG) box sequences of target genes to repress or activate their expression in the development of respective floral organs. [98]. According to the Floral Quartet Model (FQM), two dimers of the quartet recognise two respective *CArG* boxes and bind the DNA, looping it around the complex [97].

Though the binding of *CArG* domains by transcription factors forms the views of the canonical model, recent discoveries point towards specificity being conferred to MADS-box transcription factors by cofactors from different protein families, binding to different recognition sequences in the DNA. Depending on the context, MADS-box transcription factors can activate or repress transcription. They bind to a variety of corepressors or activators to perform and fine-tune this function. Comparisons are routinely drawn between MADS-box proteins and the homeodomain proteins that control the identity of cells along an axis, studied in Drosophila. Hox proteins change binding specificity according to interactions with TALE protein cofactors and it has been proposed that the recruitment of cofactors could be a strategy that holds true for MADS-box transcription factors, which also interact with TALE proteins [99,100,101,102]. It is also worth noting that interactions of these homeobox genes and their contributions to identity could be even less directly linked to the ABC model. BELL1 (BEL1) (a homeodomain protein), BR-ENHANCED EXPRESSION 1 (BEE1), HECATE1 (HEC1), and SPT were demonstrated to have a high clustering coefficient, suggesting that they may act in complexes to perform function(s) in gynoecium development, given SPT’s key role in regulating gynoecium identity [103,104,105,106].

A paper from Herrera-Ubaldo et al. [103] demonstrated that MADS-box transcription factors could be co-expressed with homeodomain transcription factors, suggesting potential protein–protein interactions that could facilitate fine-tuning gynoecium development. Homeodomain proteins were identified to be amongst the most enriched as a family in terms of their interactivity as a protein family with the MADS-box proteins [103,107]. A study by Smaczinak et al. [108] further revealed that the most prominent protein family potentially interacting with the MADS-box family in the aforementioned quaternary complexes is the homeobox family. The homeodomain protein BEL1-LIKE HOMEODOMAIN 1 (BLH1) binds to the A-class MADS domain protein AP1, as well as with ovule-specific regulators in gynoecium development [102]. In tomato (*Solanum lycopersicum*), the homeodomain protein LeHB-1 was implicated in flower development through interactions with the tomato MADS-box gene Le-MADS RIN [109]. Notably, these homeodomain protein–MADS-box interactions are not only documented in plants; they are also shown in yeast, with instances of PHOX1 homeodomain protein homologues (from humans) interacting with MADS-domain protein MCM1 in this yeast system to control the progression of the cell cycle [110]. It is evident here that there are these key interactions between identity genes and homeodomain proteins. Homeodomain proteins, as has been previously described, act to specify symmetry and axiality [63,111]. Therefore, more research should be conducted to explore whether or how homeobox genes could act in coordinating these processes. What is known will be discussed in the following section, which aims to answer: how is axiality accounted for in developing the shape of an organ?

## 4. ‘Organ’isation across the Axes

### 4.1. Shaping Identity: How Regulation of Axis Identity Informs the Identity of Floral Organs

The axes will be considered in this review in terms of apical–basal, medial–lateral and adaxial–abaxial (Figure 1, Figure 3 and Figure 5). The MADS-box genes are a relatively small group of genes conferring organ identity and therefore, as discussed, they cannot do this independently. In the pre-patterning of the floral primordia, auxin is a necessary signal. The ABCDE model functions following initiation of the organs to specify distinct identities, and genetically the two processes of initiation and identity establishment can be uncoupled [112]. This uncoupling can be further supported by the spatial distinction between auxin efflux and response maxima, which occur in the epidermis, whilst organogenesis events tend to occur in the underlying layers [38,113,114]. A further contributor to the patterning of organs is suggested by the phytomer theory. This postulates that patterning of the floral organs occurs according to adaxial and abaxial polarity across the floral primordia [115]. The development of flowers was suggested to switch from adaxial–abaxial pre-patterning of organ sites of initiation towards the well-established centripetal patterning displayed by the MADS-box genes in *Arabidopsis*. However, the importance of adaxial–abaxial polarity instruction can be seen in many leguminous species, in which initiation of organs occurs along the adaxial–abaxial body axis [116]. Further, the floral meristem forms in the adaxial axil of a cryptic bract. This adaxial initiation establishes abaxial–adaxial polarity throughout flower development [115,117]. As discussed earlier, the *pan* mutant results in a radial symmetry imposed on the *Arabidopsis* flower, where the organs assume a pentameric arrangement. This is due to the disruption of the adaxial–abaxial founder cell populations which results in the development of supernumerary sepals [34,38]. In a similar manner to leaves, the coordination between adaxial–abaxial and medio-lateral/marginal tissues plays fundamental roles in shaping the flower: it has been shown that elaboration of the floral body plan in an adaxial–abaxial-specific manner is reliant on the marginal/medial regulators *PRS/WOX3* function demonstrating the intertwined roles of axes and identity during flower organ development [38].

### 4.2. The Role of Phytohormones in Establishing Axes

From early embryogenesis in plants, the apical–basal polarity axis is determined by flux of the phytohormone auxin [118]. Auxin is generated at the basal side of two asymmetric cells dividing the zygote, and is transported apically, where a response occurs, by a series of polar auxin transporters called PINs. This is followed by a reversal in the movement of auxin toward the basal direction in the post-embryonic stage [113,119]. Interactors with the establishment of auxin patterning within the embryo are homeobox transcription factors WUSCHEL-RELATED HOMEOBOX1 (WOX8), WOX9, and WOX2. These are transcription factors that localise to the basal and apical ends of the developing zygote, respectively. WOX8/9 are activated and act in tandem with WRKY2 in order to define polarity at this stage of embryo development. WOX2 acts on PIN1 expression to control directional auxin transport, polarising it to the root pole. In this way, homeobox genes and the phytohormone auxin act together to establish this early semblance of polarity [120].

Originally, it was thought that the establishment of the leaf adaxial domain was due to the propagation of what it is called the ‘Sussex signal’ from the shoot apical meristem to the developing leaf primordium in order to direct apical–basal patterning [121]. Since then, the hypothesis of this signal has been replaced by evidence of polar auxin transport from the adaxial to the abaxial domain of the developing leaf. Auxin converges in distinct points in the developing primordium and it has been shown through incision experiments that blocking these points of convergence causes radialisation of the leaf: the loss of this adaxial–abaxial polarity [122]. To form the lamina of the leaf, it appears the marginal domains are required through the marginal homeobox genes *WOX1* and *PRESSED FLOWER* (*PRS*). Loss of these genes results in radial, abaxialised leaves, evidenced by expression of the abaxial regulator *FILAMENTOUS FLOWER* (*FIL*) throughout the structure; this demonstrates that the margin and medial domains are required to suppress the abaxial domain in plant organ shape development [123,124]. The hypothesis stands that lamina outgrowth is reliant on the careful coordination of adaxial–abaxial and medial–lateral axis specification [125].

### 4.3. The Functional Importance of Axis Specification

In the development of an organ structure, growth and the commitment of distinct cell fates across axes must be considered. Specification of the apical–basal axis is integral at all levels for the plant to distinguish what should grow above and below ground. As has been previously discussed, the axes involved are considered apical–basal, medial–lateral, and adaxial–abaxial. These axes are often distinct in function and cellular identity across tissues and their specification is critical to the shape and overall function of an organ. For instance, the basal cells of *Arabidopsis* petals elongate during development to push the petal out of the opening bud, whereas the apical specialized conical cells do not expand. These apical cells also lack the chlorophyll present at the basal cells, having redifferentiated into leukoplasts [126]. This distinction between cells at the basal part of the petal and the apical part has been studied in terms of its functionality in *Hibiscus trionum*, using the cuticular and pigmentation properties of the petals. These petals are pigmented with anthocyanin, and have a striated cuticle at their base, whereas apically they are white with conical light-scattering cells [127]. The order of these striations have been further linked to pollinator preference, with bumblebees recognizing striations with higher degrees of disorder [128].

As previously discussed, the medial–lateral axis is important to consider in the development of floral organs. *PRS* is a homeobox gene integral to the development of marginal or lateral identity in the development of organs. Where this gene was lost the development of lateral sepals was altered, as well as that of the sepal marginal cell files, and it appears that it is reliant on communication with the adaxial–abaxial axis as well [129]. Similarly, *phantastica* (*phan*) in *Antirrhinum* regulates the lateral outgrowth of leaves and petals, the loss of which occasionally results in pin-like structures forming in place of laminae [130].

At the tissue level, adaxial and abaxial identity are integral to cellular distinctions and roles. For instance, the adaxial cells in the petal are conical, optimising colour intensity to both attract pollinators and increase their grip on the flower [131,132,133,134]. However, the abaxial cells are more spherical and flat, often functioning as a site of scent production—see Figure 5A [131,135]. The sepals have similar cellular distinctions dependent on their axis. For instance, the cells at the sepals’ adaxial epidermis are smooth and rectangular, and approximately uniform in size. However, abaxially there are giant cells, which potentially play a key role mechanically in enforcing the characteristic curvature to protect the inner organs [136]. To support the argument of the coordination of axiality and identity, MADS-box genes are also found to have cellular distinctions across axes in the sepals. Through the development of the sepals, the *SEPALATA3* E-class gene expression decreases in expression abaxially, remaining adaxially expressed [137]. Thus, they may interact with specifiers of adaxial identity. Further *SEP1-4* specify adaxial sepal identity. It has been suggested that *SEP3* could play a role in patterning the axiality of these organs and therefore could be interacting with the master regulators of axiality within the floral organs [69,137].

### 4.4. The Role of Homeobox Genes in Specifying Axis and Plant Organ Shape

Homeobox genes are key genetic regulators of axiality in plant development and connect polarity, identity, and shape. An important example of this is in the patterning of the apical region of the *Arabidopsis* embryo and the development of the cotyledons. This patterning process is dependent on the activity of HD-ZIP class-III transcription factors, members of the homeobox family [138]. Homeobox genes share few conserved processes throughout the major eukaryotic kingdoms, despite their homology. They have homeotic functions regarding organ identity in Drosophila and specify the development of a body plan [12]. They do not have this same role, acting as master regulators of organ identity, in plants. However, their spatial regulation of axis is functionally conserved, and as has been previously detailed, they may interact with MADS-box proteins, which do specify organ identities in flowers. These transcription factors, including PHABULOSA (PHB), REVOLUTA (REV), PHAVOLUTA (PHV), ARABIDOPSIS THALIANA HOMEOBOX 8 (ATHB8), and ARABIDOPSIS THALIANA HOMEOBOX 15 (ATHB15), function by regulating auxin transport [139]. These *HD-ZIPIII* genes also act in adaxial patterning of cotyledons; *phb phv rev* mutants exhibit radialised cotyledons and differences in their vasculature [138]. Similarly, HD-ZIPII transcription factors, including the two transcription factors *HAT3* and *ATHB4*, previously referenced in this review, play key roles in organ patterning, with a distinct adaxial expression pattern in organogenesis. These genes have been described in terms of their role in the shade-avoidance response [140], where they promote growth under shady conditions, specification of carpel margins [141], and leaf polarity [111].

Master regulators of the adaxial–abaxial body axis are directly targeted by B-class proteins in *Arabidopsis*, demonstrating a clear interplay between MADS-box transcription factors and axial cues [142]. When considering axiality, we are, therefore, often drawn to the homeobox genes and their core role in the control of the adaxial axis. These genes coordinate development across axes specifying tissues in the adaxial to abaxial direction and so help to coordinate the development of primordia.

### 4.5. How Axiality Informs Identity and Symmetry in the Development of Floral Organ Structure

The symmetry of the flower itself, in terms of the arrangement of organs, is directed by axial-specific expression of genes, such as *CYC* in *Antirrhinum* [143]. *CYC* connects symmetry at the organ level, with its instruction of petal shape, to the overall flower level, instructing arrangement of organs [24]. Further examples demonstrate how the direction of symmetric growth within organs requires instruction across axes.

The effects of axiality on symmetry can further be seen in Figure 5B,C where loss of adaxial–abaxial regulation leads to a switch in symmetry from the bilateral wild-type leaf structure to the radialised, abaxialised leaf structure (Figure 5B). This is seen in the mutant plant *hat3athb4*, where the leaves radialise and no longer conform to the bilateral, wild-type structure. These HD-ZIPIIs adapt to changes in light in their environment; therefore, their influence on leaf shape is evidently functional [111,144,145]. A similar phenotype can be observed where the abaxial domain is lost due to loss of function of *YABBY* genes along with their co-repressors; the bilateral planar leaves will also radialise [146]. This can be seen in flower organs also, where dominant mutation of the adaxial regulator, *phb-1d*, leads to adaxialised, radial structures of both petals and sepals (only petals shown in Figure 5C) in place of their bilateral, wild-type forms. It has also been shown that the characteristic adaxial cell types form along the radialised petal structures. These adaxial cells are, as previously described, conical and easily distinguishable from their abaxial counterparts [147]. The careful coordination between adaxial and abaxial axes to form symmetrical structures can also be seen at very early stages, see Figure 5D. For instance, Peng et al. [148] showed that accelerated abaxial growth during the development of the leaf primordium forms a bilateral characteristic curvature towards the meristem, whereas in flowers, accelerated adaxial growth rates, with a further balance between the two axes, forms a radialised structure. Therefore, we show that axiality influences the symmetrical shape and function of an organ. Within the gynoecium, it is considered that the transition from bilateral to radial symmetry occurs in stages dictated by different axial coordinators across the three main axes: apical–basal, medial–lateral, and adaxial–abaxial. In the first instance, where the ovary first begins its transition, shouldering the formation of the radial style, the medial regulators *SPT* and *IND* coordinate the triggering step. Following this, the adaxial coordination comes into play with *SPT* again, this time with *HECs*, supporting the transition towards radiality (Figure 1 and Figure 3). Downstream of these genes, the expression of adaxial regulators *HAT3* and *ATHB4* is promoted, consolidating this role of axial transitions and supporting final apical fusion of the carpel in a radial, solid structure, efficient to support fertilization. In order to do this, these genes must interact with and coordinate auxin transport, facilitating the progression auxin efflux from two focal points towards an auxin ring. These adaxial regulators also act to regulate the expression of the gynoecium master regulator SPT. In the double mutant line of these genes, *hat3athb4*, *SPT* expression is increased within the gynoecium [55,63]. This therefore draws a direct link between axial and identity regulators in the development of floral organ shape.

Therefore, this supports a hypothesis that homeodomain transcription factors could act via conserved mechanisms to coordinate the processes of symmetry establishment and the determination of axiality and identity. This could either be genetically or through direct interaction (in tetrameric complexes) with identity and symmetry regulators. Whether and how this is the case needs to be investigated with future research but could provide exciting insights into the genetic network and biochemical dynamics underlying flower development.

## 5. Conclusions

The genetic network of specification through to shape emergence and formation has yet to be fully and comprehensively understood. This has immense and critical potential for furthering our understanding of fruit and seed production as well as pollination to progress agriculture. This review has demonstrated that we have a promising foundation on which we can build our investigation into the coordination of fundamental processes in floral development. Research into identity was transformed in 1991, and we began to understand how these floral identity regulators are homeotic in the same way as homeobox genes in animals [8,9]. However, interactions between these two sets of genes are not well understood. Further work in this area will allow us to understand, in a spatial and temporal manner, how floral organ identity interacts with axiality to form symmetric arrangement within and of floral organs. We propose a mechanism whereby these coordinators are highlighted in the development of symmetry within the gynoecium, and may act via a conserved mechanism to coordinate these processes in other floral organs, allowing for further insight into the development and organisation of shape in general. In this way, this review posits a potential mechanism whereby the homeotic genes controlling organ identity in a spatio-temporal manner (the MADS-box genes controlling floral organ identity) associate with genes known to spatio-temporally regulate identity across axes in plants to establish and form the appropriate structures, potentially resolving distinct processes and networks in flower development [8,63].

## Figures and Tables

**Figure 1 plants-13-01595-f001:**
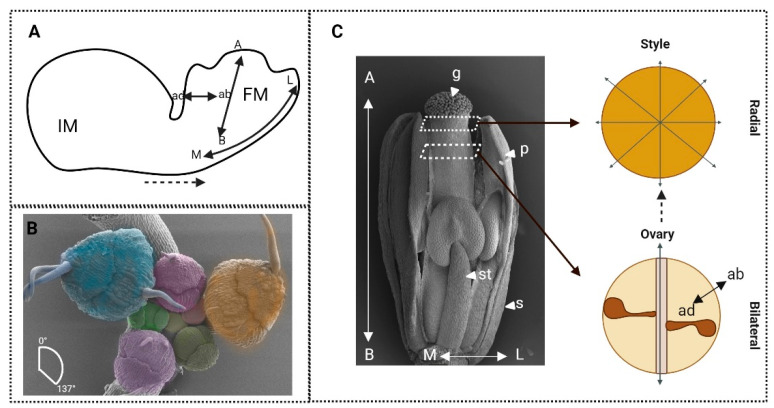
The development of the complex shape of a flower is reliant on the combined participation of developmental processes specifying axis growth, identity specification, and symmetry organisation. (**A**) The transition from a spherical inflorescence meristem (IM) to a floral meristem (FM) and a developed flower requires coordinated growth across adaxial (ad), abaxial (ab), apical (A), basal (B), and medial (M), and lateral (L) axes. (**B**) Scanning electronic micrograph showing the phyllotactic arrangement of floral buds (differently false-coloured) and primordia at roughly 137° of an *Arabidopsis thaliana* inflorescence. (**C**) Scanning electronic micrograph of a mature *Arabidopsis* flower at stage 12 of its development, showing sepals (s), petals (p), stamens (st), and the central gynoecium (g) (note, some sepals and petals have been removed to show organs located in the central whorls). Schematic transverse sections (dashed lines) at the style and ovary regions of the gynoecium displaying radial and bilateral symmetry, respectively.

**Figure 2 plants-13-01595-f002:**
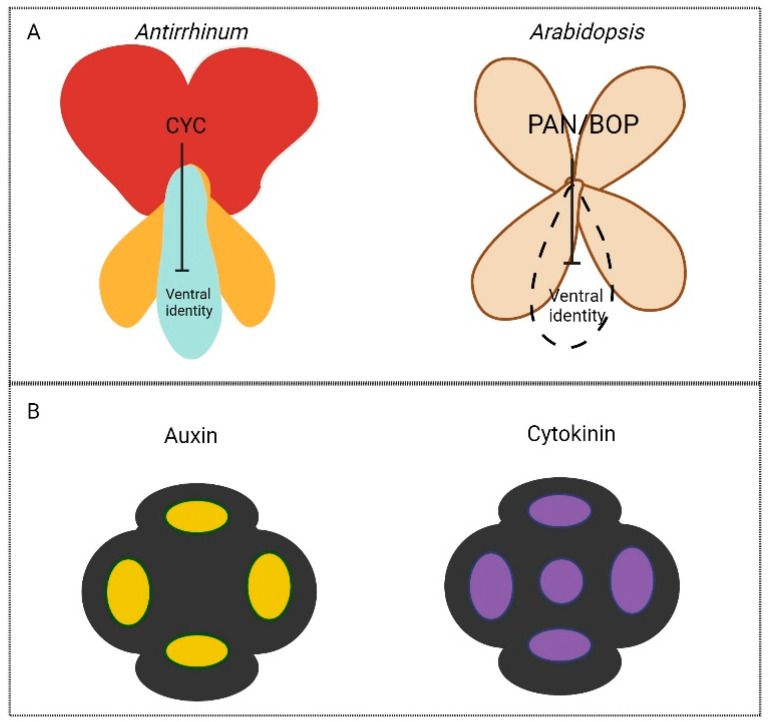
The initiation of floral organs is coordinated through inhibitory genetic mechanisms, spatial coordination, and the activity of plant hormones. (**A**) Inhibition of ventral identity in petals by *CYC* (*Antirrhinum*) and *PAN/BOP* (*Arabidopsis*). The dorsal petals (red) are a distinct shape to the lateral (yellow) and ventral (blue) petals in *Antirrhinum*, whereas in *Arabidopsis*, all petals are the same (brown). (**B**) Auxin and cytokinin signalling output (yellow and purple, respectively) pattern sites of sepal organ initiation in *Arabidopsis* floral primordia (black outline).

**Figure 3 plants-13-01595-f003:**
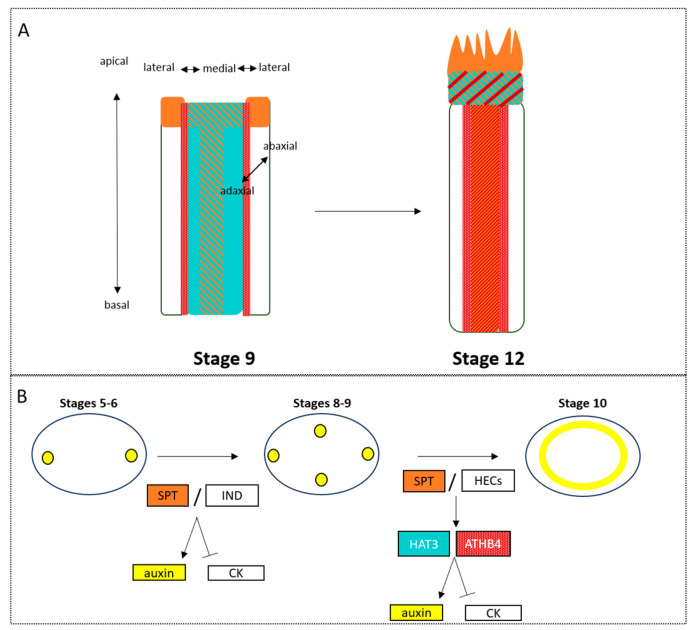
The development of the female reproductive organ, the gynoecium, in *Arabidopsis*, requires step-wise recruitment of axis and auxin distribution. (**A**) The expression patterns of *HAT3* (cyan), *ATHB4* (red), and *SPT* (orange), coordinators of axial development, in stage 9 and stage 12 gynoecium. (**B**) The stepwise development of the style from stage 5 to 10, in terms of auxin localisation (yellow) as regulated be the activity of the transcription factors shown in the figure.

**Figure 4 plants-13-01595-f004:**
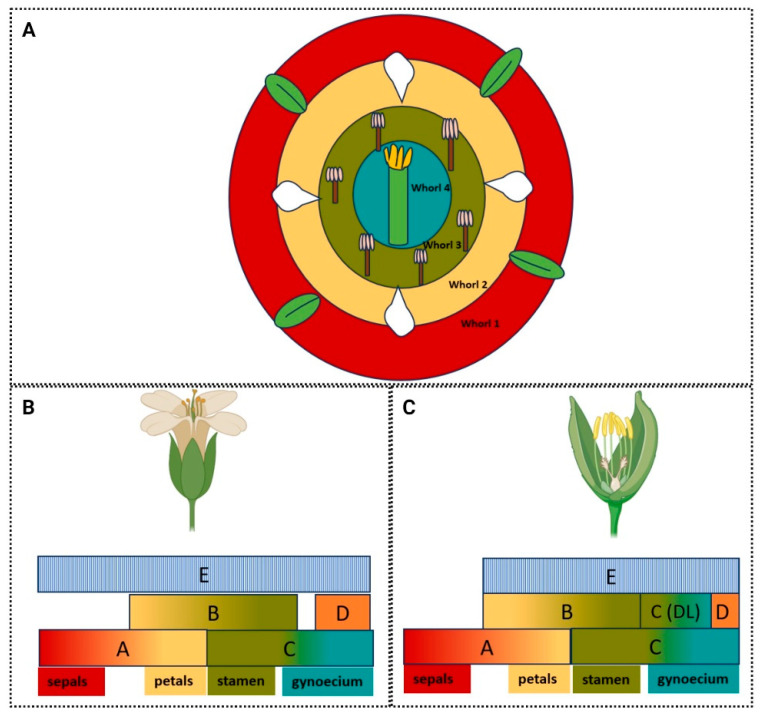
The ABCDE model describes the arrangement of floral organs by classes of genes acting as master regulators of organ identity. (**A**) The concentric whorls of an *Arabidopsis thaliana* flower, enclosing the organs they define: the gynoecium, stamen, petals, and sepals from the innermost whorl extending outwards. (**B**) The ABCDE model as described in the dicotyledonous Arabidopsis flowers. (**C**) The ABCDE model in the monocotyledonous rice model flower; DL stands for the DROOPING LEAF, the homeotic equivalent to the C-class gene in defining carpel identity. Sepals in whorl 1 are specified by A class genes (red), petals in whorl 2, are specified by A and B class genes (yellow), stamen in whorl 3 are specified by B and C class genes (green) and the gynoecium in whorl 4 is specified by the C class genes (blue).

**Figure 5 plants-13-01595-f005:**
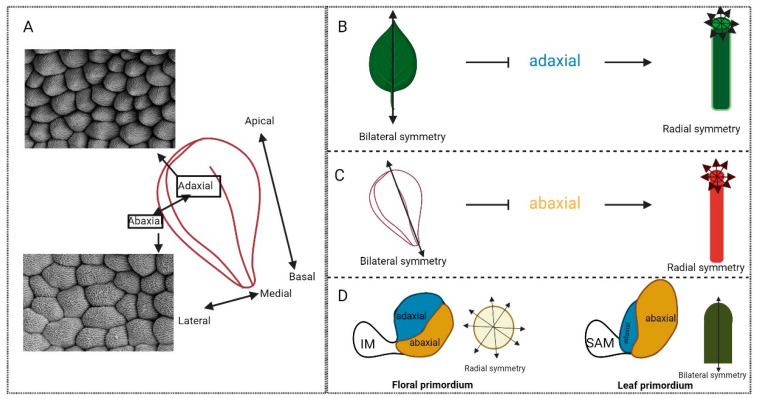
The role of axiality at the tissue and organ level to direct the symmetry and function of organs. (**A**) Schematic representation of a petal organised across axes with scanning electron micrographs of abaxial and adaxial cell characteristics. (**B**) Schematic representation of a bilateral (**left**) and radial (**right**) leaf. Loss of adaxial identity in a leaf system results in radialisation of the leaf. (**C**) Schematic representation of a bilateral petal where loss of abaxiality through imposed adaxiality has the same effect: radialisation of the petal structure. (**D**) Schematic representation of how coordinated growth rates across the adaxial–abaxial axes form differing symmetries in the establishment of the floral and leaf primordia.

## Data Availability

No new data and resources were produced to complete this review.

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
