# Peer review of "‘Organ’ising Floral Organ Development"

_plants, 2024, doi:10.3390/plants13121595_

Round 1

Reviewer 1 Report

Comments and Suggestions for Authors

MS ID: plants-3015047

The authors provided an extensive review about the floral symmetry. They mentioned that syymmetric shapes, colors, and scents all play important functional roles in flower biology, which are evolved from leaves to specialize in distinctive whorls functioning as reproductive organs. They showed that the evolution of flower symmetry and the morphology of individual flower parts, i.e. sepals, petals, stamens, carpels, has significantly contributed to the diversity of reproductive strategies across flowering plant species, which facilitates attractiveness for pollination, protection of gametes, efficient fertilization, and seed production. Complex genetic network underlies the establishment of organ, tissue, and cellular identity with growth regulators acting across the body axes. They said that we summarized the wealth of research together to show how separate processes interact in specifying organ identity within the flower, and provided a functional perspective on how identity determination and axial regulation are coordinated to inform symmetrical flower organ structures. They also said that they proposed a mechanism whereby these coordinators highlighted in the development of symmetry within the gynoecium, may act via a conserved mechanism to coordinate these processes in other floral organs, allowing for further insight into the development and organization of shape in general. They especially emphasized that the homeotic genes, controlling organ identity in a spatiotemporal manner (the MADS-box genes controlling floral organ identity), are associated with genes known to spatiotemporally regulate identity across axes in plants to establish and form the appropriate structures, potentially resolving distinct processes and networks in flower development.

However, there are too many narrative contents of the ABCDE model and related MADS-box genes in this manuscript. MADS-box genes in the ABCDE model are mainly related to floral organ identity rather than the floral symmetry. The two kinds of transcription factors (MADS-box and TCP) may have interaction in shaping each whorl of flowers in floral symmetry, but the key factor controlling the floral symmetry is TCP transcription factor interacting with other regulators, such as DIV, RAD (MYB transcription family). In addition, the title of this manuscript “The diversity of floral symmetry: ‘Organ’ising floral organ development” is not appropriate because the manuscript was not developed in a relevant manner to the diversity of floral symmetry. Furthermore, there are frequent conceptual errors occurred in their manuscript, such as “CYC acts ventrally in Antirrhinum”. The gene CYCLOIDEA in snapdragon essentially acts dorsally in floral organs with no action in the ventral organs.

Author Response

Reviewer 1

The authors provided an extensive review about the floral symmetry. They mentioned that syymmetric shapes, colors, and scents all play important functional roles in flower biology, which are evolved from leaves to specialize in distinctive whorls functioning as reproductive organs. They showed that the evolution of flower symmetry and the morphology of individual flower parts, i.e. sepals, petals, stamens, carpels, has significantly contributed to the diversity of reproductive strategies across flowering plant species, which facilitates attractiveness for pollination, protection of gametes, efficient fertilization, and seed production. Complex genetic network underlies the establishment of organ, tissue, and cellular identity with growth regulators acting across the body axes. They said that we summarized the wealth of research together to show how separate processes interact in specifying organ identity within the flower, and provided a functional perspective on how identity determination and axial regulation are coordinated to inform symmetrical flower organ structures. They also said that they proposed a mechanism whereby these coordinators highlighted in the development of symmetry within the gynoecium, may act via a conserved mechanism to coordinate these processes in other floral organs, allowing for further insight into the development and organization of shape in general. They especially emphasized that the homeotic genes, controlling organ identity in a spatiotemporal manner (the MADS-box genes controlling floral organ identity), are associated with genes known to spatiotemporally regulate identity across axes in plants to establish and form the appropriate structures, potentially resolving distinct processes and networks in flower development.

However, there are too many narrative contents of the ABCDE model and related MADS-box genes in this manuscript. MADS-box genes in the ABCDE model are mainly related to floral organ identity rather than the floral symmetry. The two kinds of transcription factors (MADS-box and TCP) may have interaction in shaping each whorl of flowers in floral symmetry, but the key factor controlling the floral symmetry is TCP transcription factor interacting with other regulators, such as DIV, RAD (MYB transcription family).

Response: We agree with the reviewer’s comment. However, our narrative contents about the ABCDE model serve to highlight their role for identity determination, and how that can be coordinated with the establishment of the body axis, thus providing an alternative scenario for symmetry foundation during flower organ morphogenesis.

In addition, the title of this manuscript “The diversity of floral symmetry: ‘Organ’ising floral organ development” is not appropriate because the manuscript was not developed in a relevant manner to the diversity of floral symmetry.

Response: We have removed “The diversity of floral symmetry” from the title.

Furthermore, there are frequent conceptual errors occurred in their manuscript, such as “CYC acts ventrally in Antirrhinum”. The gene CYCLOIDEA in snapdragon essentially acts dorsally in floral organs with no action in the ventral organs

Response: We mistakenly swapped the dorsal-ventral terms and apologise for this involuntary mistake. We have corrected the text accordingly (see line 130).   

Reviewer 2 Report

Comments and Suggestions for Authors

This review considers the regulatory pathways controlling floral organ development, linking recent findings to build general arguments to guide future progress of the field. This review is of interest to the broad field of developmental biology.
The abstract summarises the topic of the review very well, highlighting the evolutionary and functional importance of flower development and setting out the timeliness of the review to bring together recent findings in the field. The introduction starts with further background highlighting the importance, history, and humanities aspects of the system. Carefully designed figures help illustrate key spatial concepts. The main discussion is structured well to build arguments through the sequence of developmental stages required to generate flower form, drawing on a large body of relevant literature. Some nice analogies are drawn with animal systems at various points. Short reminders at the begninning of each section are useful to follow the progression of the arguments. The conclusion summarises the main arguments well with a prospectus for future research. Overall is the a very complete review of the current state of the field, that builds some interesting hypotheses to further influence the field's development.

Specific comments
L84-86 This is rather a jump of logic. Perhaps there are examples of symmetry being beneficial in plant/pollinator systems that can be referenced as well.
L94-98 This paragraph is covers similar ideas to the last but would be a good example for the previous comment.
L121-124 Figure 2A helps explain the example but a little more detail would help here in the text.
L158-161 Specify if the processes described here apply to the top or bottom of the gynoecium.
L164 A 45 degree rotation of the floral diagrams in panel B would match better with the arabidopsis flower model shown in panel A.
L231 There seem to be two shades of orange in  Figure 3 A and B. Identify both in the legend if they are meant to be different.
L278 Switch "chemically" to "biochemically"?
L360 "others" to "others'"
L389-390 Specify if this protein has a homeodomain according to the argument being described.
L566-571 Check the grammar of this statement; there seems to a misisng "or" clause.

Author Response

Reviewer 2

This review considers the regulatory pathways controlling floral organ development, linking recent findings to build general arguments to guide future progress of the field. This review is of interest to the broad field of developmental biology.
The abstract summarises the topic of the review very well, highlighting the evolutionary and functional importance of flower development and setting out the timeliness of the review to bring together recent findings in the field. The introduction starts with further background highlighting the importance, history, and humanities aspects of the system. Carefully designed figures help illustrate key spatial concepts. The main discussion is structured well to build arguments through the sequence of developmental stages required to generate flower form, drawing on a large body of relevant literature. Some nice analogies are drawn with animal systems at various points. Short reminders at the begninning of each section are useful to follow the progression of the arguments. The conclusion summarises the main arguments well with a prospectus for future research. Overall is the a very complete review of the current state of the field, that builds some interesting hypotheses to further influence the field's development. 

Specific comments
L84-86 This is rather a jump of logic. Perhaps there are examples of symmetry being beneficial in plant/pollinator systems that can be referenced as well.
L94-98 This paragraph is covers similar ideas to the last but would be a good example for the previous comment.

Response: To answer both comments, we have modified this paragraph by adding examples of symmetric flowers/pollinators preferences and explained why it also influences human psychology (please see lines 88-93).

L121-124 Figure 2A helps explain the example but a little more detail would help here in the text. 

Response: We have explained better the role of PAN in regulating flower organs number (please see Lines 118-125).

L158-161 Specify if the processes described here apply to the top or bottom of the gynoecium.

Response: We have clarified that in the text (please see Lines 164-169).

L164 A 45 degree rotation of the floral diagrams in panel B would match better with the arabidopsis flower model shown in panel A.

Response: In the figure caption of Fig. 2, we have specified that panel A shows petal initiation in panel A and sepal initiation in panel B. Because sepals arise alternately to petals, hence the meristem outline is at a 45° reorientation to the apical view of the petals in A.

L231 There seem to be two shades of orange in  Figure 3 A and B. Identify both in the legend if they are meant to be different.

Response: The shades of orange have been corrected to suit this comment, and make it easier for the reader to identify the expression pattern of the corresponding genes.

L278 Switch "chemically" to "biochemically"?

Response: We added both chemically and biochemically and provided two references (please see Line 281).

L360 "others" to "others'"

Response: The sentence has been modified as follows: “[…] in regulating flower development is reliant from other organ inputs” (please see Line 366).

L389-390 Specify if this protein has a homeodomain according to the argument being described.

Response: Done (please see Line 401).

L566-571 Check the grammar of this statement; there seems to a misisng "or" clause.

Response: Done (please see Line 590).

Reviewer 3 Report

Comments and Suggestions for Authors

Dear authors

The Ms is a comprehensive review. I made few comments and suggestions in the pdf that I would like you to take into consideration.

Additionally, I noticed that Laila Moubayidin cited 18 imes articles where is an author. The majority of the other authors are cited 2 or 3 times and many auhtors are cited only one time. Could you please consider this aspect?

Kind regards.

Author Response

Reviewer 3

Dear authors

The Ms is a comprehensive review. I made few comments and suggestions in the pdf that I would like you to take into consideration.

Comment: in some cases the edible part is the seed

Response: We have added “seeds” to the list (see line 26).

Comment: this demand is a consequence of the population growth. It would be adviceable to clarify the origin of the demand.

Response: We have modified the text accordingly by adding “the demand of a growing population” (see line 29).

Comment: a very important reference is missing: Saedler and Huijser 1993 (doi: 10.1016/0378-1119(93)90071-a), which referes to genes involved in the determination of the floral meristem. Mutations in squamosa and floricaula genes lead to bracts where flowers would be expected. Therefore, in these cases, there are no flowers. This should be referred.

Response: We have added the reference mentioned above and specified the role of FLORICAULA (FLO) and SQUAMOSA (SQUA) in the model organism Antirrhinum majus (see lines 40-43).

Comment: and Antirrhinum majus. Several studies on A. majus should be used in this review

Response: We have added Antirrhinum majus (see line 47) and added further studies on A. majus starting from lines 151, 332, 526.  

Comment: This statment is not in line with the title. Focusing only on A. thaliana does not reflect the existing variability of floral symmetry. I suggest to change the title or to refer other flower model systems as you do in the subsequent chapters

Response: We have removed “The diversity of floral symmetry” from the title and modified the text by referring to the plant kingdom (see lines 53-59).

Comment: No examples of A, B and C genes are given, whereas for D and E classes several genes are mentioned. Why didn't you refer genes as Def, Glo, Choripetala, Fimbriata or Despenteado,among others?

Response: We have added the names of A,B and C genes from A.thaliana (see lines 292-297) and referred to DEFICIENS (DEF)and GLOBOSA (GLO) in the following section (Diversity of the ABCDE model) (see lines 333-336). We did not include the other players mentioned by the reviewer as we believe it draws away from main point of conservation of genes and mechanisms.

Additionally, I noticed that Laila Moubayidin cited 18 imes articles where is an author. The majority of the other authors are cited 2 or 3 times and many auhtors are cited only one time. Could you please consider this aspect?

Response: There were points where these articles were cited twice, erroneously, in a sentence. Thank you for raising this issue, it has been amended and this number of citations has been reduced to 12. This number cannot be further reduced as the content from these papers is integral to the structure of the review.

Round 2

Reviewer 1 Report

Comments and Suggestions for Authors

The new version addresses my major concerns, so, I recommend this manuscript for publication in Plants.

Author Response

Thanks.
